# A Real-Time Detection Device for the Rapid Quantification of Skin Casual Sebum Using the Oil Red O Staining Method

**DOI:** 10.3390/s22083016

**Published:** 2022-04-14

**Authors:** Kung Ahn, Sangjin Han, Kyeongeui Yun, Wooseok Lee, Dong-Geol Lee, So Min Kang, Young-Bong Choi, Kyudong Han, Yong Ju Ahn

**Affiliations:** 1HuNBiome Co., Ltd., R&D Center, Gasan Digital 1-ro, Geumcheon-gu, Seoul 08507, Korea; kevin@hunbiome.com (K.A.); jin@hunbiome.com (S.H.); sindy@hunbiome.com (K.Y.); 2Center for Bio-Medical Engineering Core Facility, Dankook University, Cheonan 31116, Korea; wooseoklee87@gmail.com; 3Department of Microbiology, College of Science & Technology, Dankook University, Cheonan 31116, Korea; leedg@cosmax.com; 4R&I Center, COSMAX BTI, Pangyo-ro 255, Bundang-gu, Seoungnam-si 13486, Korea; 5Department of Plastic and Reconstructive Surgery, Seoul National University Bundang Hospital, Bundang-gu, Seongnam 13620, Korea; doctork721@koreansrc.com; 6H&BIO KSRC (H&BIO Korean Skin Research Center), Bundang-gu, Seongnam 13605, Korea; 7Department of Chemistry, College of Science & Technology, Dankook University, Cheonan 31116, Korea; chem0404@dankook.ac.kr

**Keywords:** ORO (Oil Red O), facial casual skin sebum, sebum measurement, color sensor TCS3200, Arduino system

## Abstract

The human skin sebum suggests that it (along with other epidermal surface lipids) plays a role in skin barrier formation, the moderation of cutaneous inflammation, and antimicrobial defense. Various methods have been developed for collecting and measuring skin sebum. We tested methods of detection using “color intensity”, by staining the skin casual sebum. This process was conducted in three steps; first, the selection of materials for sebum collection; second, staining the collected sebum; third, the development of a device that can measure the level of stained sebum. A plastic film was used to effectively collect sebum that increased with the replacement time of the sebum. In addition, the collected sebum was stained with Oil Red O (ORO) and checked with RGB; as a result, the R^2^ value was higher than 0.9. It was also confirmed that the correlation value was higher than 0.9 in the comparison result with Sebumeter^®^, which is a common standard technology. Finally, it was confirmed that the R^2^ value was higher than 0.9 in the detection value using the sensor. In conclusion, we have proven the proof of concept (PoC) for this method, and we would like to introduce an effective sebum measurement method that differs from the existing method.

## 1. Introduction

Sebum is an oily and waxy substance secreted by the sebaceous glands of the skin, and represents one of the two main cutaneous secretions (the other being sweat). Sebum is secreted throughout the body (with the exception of the palms of the hand and soles of the feet), and its production has been reported to be most prominent in the forehead, nose, and chin (the so-called “T-zone” of the face). The specific functions of sebum are yet to be fully elucidated, but the composition of sebum suggests that it (along with other epidermal surface lipids) plays a role in the maintenance of skin barrier function, moderation of cutaneous inflammation, and antimicrobial defense against harmful microbes [1,2,3,4]. Over-secreted sebum has also been reported to be associated with inflammatory skin conditions, such as acne [5]. Various methods have been developed for collecting sebum, such as solvents [3], cigarette paper [4], bentonite gel [6], frosted glass [7], and plastic films [8]. In addition to collecting skin sebum, technologies have been developed to measure quantitative amounts. Among these technologies, the Sebumeter^®^ (Courage + Khazaka electronic GmbH, Cologne, Germany) is the most widely used instrument for sebum measurement. The Sebumeter^®^ is a well-known device for quantitatively detecting the amount of sebum using an optical method. In particular, this method uses a film to collect skin sebum [8,9,10].

In this study, we conducted methods for detection using “color intensity”, by staining the skin casual sebum in three parts for effective measurement. The first is the selection of materials for sebum collection; the second is the staining of the collected sebum; and, finally, the development of a device that can measure the level of stained sebum in real time in a cost-effective way. First of all, we performed staining using various commercial blotting papers and plastic films for collecting the skin sebum. We then selected Oil Red O (ORO) as a reagent for staining the skin sebum. ORO, also called solvent red 27, is a lipophilic dye that stains fat and lipid components in biological samples [11,12,13,14,15,16,17,18]. It is also used to stain oil and wax red. ORO absorption has a wavelength of 518 nm and stains the item red [17]. Oil Red O staining is simple, fast, cost effective, and non-destructive, and requires less equipment to process items [11,12,13,14,15,16,17,18]. Therefore, we selected the ORO staining method for the quantification of casual sebum in this study. Finally, we further developed Arduino and a color sensor-based device for the successful detection and quantification of casual sebum using PVC films. Arduino is an open-source electronics platform based on easy-to-use hardware and software (https://www.arduino.cc/), which can be used to design an infinite number of devices with different properties. It is also attracting a lot of interest in various scientific communities [19].

Herein, we have provided the proof of concept (PoC) for this method, and we would like to introduce an effective sebum measurement method that is different from the existing methods.

## 2. Materials and Methods

### 2.1. Skin Sebum Collection and Staining

(1)Prepare the materials for collection of skin sebum

The time required to recover the casual level of sebum after cleansing the sebum from the skin surface is called the sebum replacement time [20,21]. The sebum replacement time has been reported to be 1 h~2 h in all seborrheic areas [20,21]. Commercial oil blotting papers and polyvinyl chloride (PVC) films were used for collecting facial skin sebum. Paper and films cut to 2 × 3 (cm) size areas were disinfected with alcohol. Participants were kept free of oil from our hands or other contacts (Appendix A).

(2)Prepare the Oil Red O staining solution and buffer solution

Two solutions were prepared for ORO staining (Sigma-Aldrich Corp., St. Louis, MO, USA) and a buffer solution for skin sebum staining. The ORO reagent (1.54 g) was dissolved in methanol (770 mL). A solution of sodium hydroxide (9.2 g) in water (230 mL) was prepared. An aqueous solution of sodium hydroxide was added to the alcoholic solution of ORO. The contents were then thoroughly mixed and filtered. The filtrate was stored in a dark brown bottle away from light [13]. The stain solution, which was kept in a dark atmosphere, remained stable for approximately 8 months, even though the color of the solution changed from orange to yellow after 2 months [12,13]. The buffer solution was prepared as follows: sodium carbonate (26.5 g) was dissolved in water (2.0 ℓ) and shaken until soluble. Concentrated nitric acid (70%, 18.3 mL) was added to the carbonate solution in small lots with constant stirring. The contents were diluted by adding water (2.5 ℓ) and stored in a dark-colored bottle [18].

(3)ORO staining

ORO staining was performed for causal sebum staining as follows: After wiping the forehead with cleansing tissue (2’~3’ required) and drying the “T-zone” of the forehead, the film was cut into 2 × 3 (cm) sizes, attached to the forehead, and peeled off immediately. Then, using tweezers, the contact surface was placed at the center (so that the contact surface could be stained well) of the ORO reagent, removed (3 min), washed in a buffer (3 min), and dried.

(4)Absorbance of ORO-stained sebum

Lipids were then extracted from the cells using 500 uL of isopropyl alcohol at 75 °C for 5 min. After the extraction step, 400 uL of deionized water and 400 uL of n-hexane were added to an Eppendorf tube and mixed well by vortexing for 5 s. Subsequently, the mixture was separated by centrifugation at 500× *g* for 30 s. The upper n-hexane layer was then transferred to a 10 mm acryl cuvette, and the concentration of the lipids was measured based on the absorbance at 345 nm. The absorbance was measured using a UV–Vis spectrophotometer mini 1240 model (Shimadzu, Kyoto, Japan).

### 2.2. The RGB Color Analysis

(1)Digitization of stained color data

Digital images were recorded from a smartphone for RGB analysis using ImageJ software (https://imagej.nih.gov/ij). RGB analysis of the digital images was carried out using ImageJ (1.53k), an open-source program from the National Institutes of Health and the Laboratory for Optical and Computational Sciences. To reduce noise in the data, owing to edge effects and shadows, square areas from the center of the images were selected.

### 2.3. Color Detection using Sensor

(1)Hardware and RGB color sensor

The acrylic was cut according to the size and design of the device. A schematic of the components is shown in Appendix A. The device consists of the TCS3200 color sensor, an I2C LCD, an Arduino UNO controller for control and readout of data, and a Bluetooth module (HM-10) for data transfer to the mobile application. The TCS3200 color sensor is connected to Arduino UNO for voltage supply and readouts. The respective pins 4, 5, 6, and 7 control the output frequency scaling, the set of photodiodes (red/green/blue/clear) and the LED on the front. The pins are connected to the digital output pins of the Arduino. I2C LCD and Bluetooth module HM-10 are connected using an Arduino expansion shield (Appendix A).

(2)Software and the application

The program coding was performed using the Arduino sketch. The mobile application for sebum level monitoring was developed, and it can be easily applied to skincare solutions for various purposes (Appendix A).

## 3. Results and Discussion

### 3.1. Selection of Materials for Facial Casual Skin Sebum Collection

First, four commercial oil blotting papers (pulp material) were used to collect the facial skin sebum (Figure 1), and ORO staining was carried out to detect the collected sebum. On the facial skin site, the “T-zone”, which is a well-known sebum-rich site on the facial skin, was selected. Facial casual skin sebum was collected from the T-zone using commercial oil blotting paper at 0, 30, and 60 min after facial washing. Subsequently, ORO staining was performed using oil blotting papers. However, as shown in Figure 1a, it was difficult to quantify the casual sebum level using the color level, owing to the red and pink background color noise. Therefore, we attempted to detect the sebum using PVC plastic films that can be easily stained with sebum (Figure 1b). This method of using a plastic film is typically used in the Sebumeter^®^ system, but it is a principle of measuring the amount of sebum through transparency, by placing the sebum on a transparent film [8].

As mentioned above, in consideration of the sebum replacement time, the sebum was collected at 60 min after facial washing in the “T-zone” of the facial skin in the same way. The collected sebum was stained with ORO. As a result, we could detect a difference in color level between the ORO-stained plastic film with and without sebum (non-staining). The ORO-stained plastic film with facial skin sebum showed increasing patterns until the sebum replacement time. As shown in Figure 2a, the color density is dark pink in color with sebum at 0, 30, and 60 min after facial washing, compared to “Non-staining”, in the ORO-stained plastic film. Moreover, the digitalization of the color density was converted into its RGB components using ImageJ (1.53k) (https://imagej.nih.gov/ij), to precisely confirm the color change (Figure 2b). The calculated mean RGB intensities of the ORO-stained sebum were converted to their chromaticity values x and y. The color chromaticity diagram indicates that the color of the ORO-stained sebum changed from bright pink to dark pink with an increase in the concentration of the skin sebum (Figure 2c).

To confirm the increase in sebum amount, we collected sebum according to the sebum replacement period, and performed ORO staining. Sebum was collected from individuals who had no skin diseases, and were diagnosed with dry and oily skin types by a dermatologist. Sebum was collected in the “T-zone” of the forehead after washing for 0, 30, and 60 min, using a fabricated PVC plastic film that was 2 × 3 (cm) in size. The visual change in color density on the stained sebum after facial washing is illustrated in Appendix A. The color density of the stained ORO (pink color) showed an increasing pattern according to the sebum replacement time. Digitalization of the color density was converted to RGB components using the ImageJ software, to confirm the color change. A correlation analysis between the sebum replacement time and color change was performed. Each RGB value has a maximum value of 255 in digital color. Our results confirmed that the red value was larger than the average of the green and blue values. The level of stained sebum was calculated using the following equation, and we named the “casual sebum level” in the corresponding values: Casual sebum level = Red − ((Green + Blue)/2) (1)

The R^2^ values of the casual sebum are 0.965, 0.952, 0.408, and 0.9789 from four participants by the sebum replacement time (Table 1). To confirm the amount of sebum stained with ORO, we carried out an absorbance analysis on isolated sebum. Appendix A show that the absorbance increased in accordance with the color of the sebum stained with ORO. Based on this result, the possibility of the concept can be confirmed.

### 3.2. Comparison with Sebumeter^®^

We compared our method with a common standard technology from other samples for reproducible verification. The participants in this experiment consisted of two men and one woman diagnosed with oily skin and without skin diseases. Sebum was collected from the “T-zone” by considering the sebum replacement time at 0, 30, and 60 min after facial washing. The sebum was then stained and digitalized. For comparison with our methods, the casual sebum levels were measured with a Sebumeter^®^, using the same process. The correlation between the intensity of ORO-stained color and the value from the Sebumeter^®^ was 0.946, 0.996, and 0.994 in the three samples, respectively (Table 2). Therefore, sebum collection using PVC plastic films and ORO staining for quantitative sebum measurement is expected to be an effective method for measuring casual sebum levels.

### 3.3. Detection of Facial Casual Skin Sebum Using Color Sensor

We developed a color sensor-based device to measure the level of skin sebum with a low cost and high efficiency (Appendix A). Equipment testing was performed using a TCS3200 color sensor, and changes in the ORO-stained color were analyzed by the sebum replacement time. The output of the TCS3200 sensor was recorded with its frequency. Therefore, we corrected the color value based on the frequency. The maximum value of the frequency was corrected to the maximum RGB value (255, 255, and 255). The stained color value was observed using this system. For device performance testing, samples were collected from five individuals without skin diseases.

As shown in Figure 3, the methods using a Sebumeter^®^ and the fabricated device were compared. All the experiments were performed in triplicates. Both the Sebumeter^®^ and our device showed a significant increase in the amount of sebum at the sebum replacement time. It was also confirmed that the deviation in each sample, according to the repeated experiment, was smaller in our system than in the Sebumeter^®^ (Appendix A). Overall, the plastic film was used to effectively collect the sebum that increased with the sebum replacement time. In addition, the collected sebum was stained with ORO and checked with RGB, which resulted in the R^2^ value being higher than 0.9. It was also confirmed that the correlation value was higher than 0.9 in the comparison result with the Sebumeter^®^, which is a common standard technology (Appendix A). Although this study was conducted to monitor sebum in a limited number of participants, we will monitor more participants in future studies. This system could be used to objectively monitor the amount of sebum based on a specific facial skin type (dry, oily, or complex). In this study, only the sebum on the face was measured, but sebum measurements in other parts of the body are also required. This system can be used to monitor sebum changes in various skin diseases.

## 4. Conclusions

Skin sebum is an important factor in facial skincare, because there are significant differences in facial skin types, in terms of the amount of sebum secreted. In general, skin is classified into three types (oily, normal, and dry), according to each individual’s subjective feelings concerning sebum secretion. However, individual assessments based on feelings are subjective. To objectively classify the skin types, several methods have been developed to measure sebum secretion (Appendix A). ORO is an effective and efficient method for developing lipid content staining. Although the collection of sebum using PVC plastic films has been commercialized, based on a similar concept, there is no method to detect the collected contents through staining. In addition, a method for detecting the chromaticity of sebum stained with ORO, based on a color sensor, was first attempted in this study. In summary, we successfully developed methods for the detection of casual sebum based on ORO staining. The casual skin sebum was effectively collected using PVC plastic films and stained well by ORO. Moreover, the proposed technique was applied to ORO-stained sebum coupled with a smartphone and an Arduino-based device as a detection system (Appendix A). The visual detection of the method was also convenient and cost effective in detecting casual skin sebum.

## Figures and Tables

**Figure 1 sensors-22-03016-f001:**
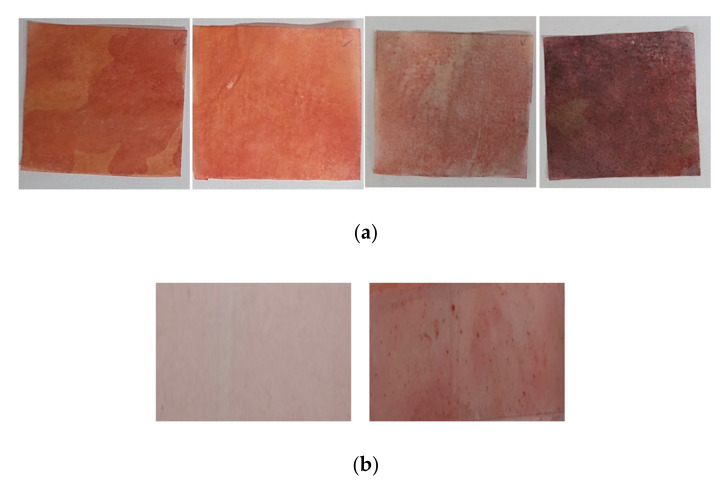
ORO staining test for facial skin sebum collection in commercial oily papers and PVC plastic films. (**a**) ORO staining using four commercial oil blotting papers. (**b**) The left figure presents a plastic film after Oil Red O staining without sebum, while the right figure depicts a plastic film after sebum staining with ORO.

**Figure 2 sensors-22-03016-f002:**
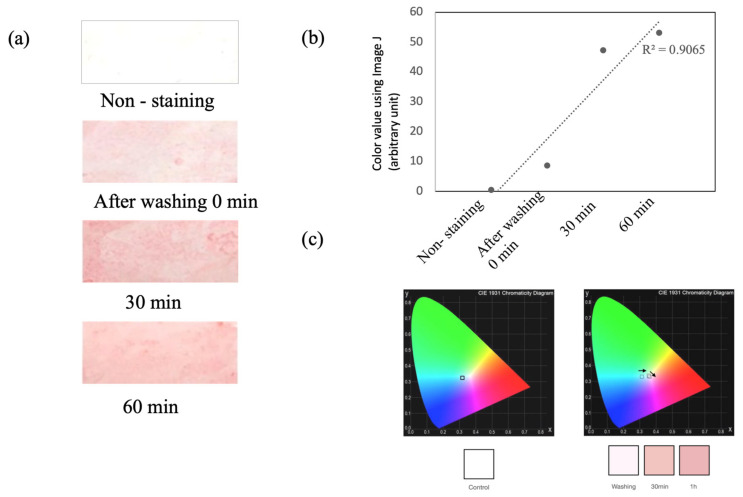
Correlation between the sebum replace time and ORO-stained color intensity. (**a**) ORO-stained sebum color intensity collected from PVC film by sebum replacement time. (**b**) The coefficient of determination from digitalized color value using ImageJ. (**c**) CIE diagram from chromaticity values.

**Figure 3 sensors-22-03016-f003:**
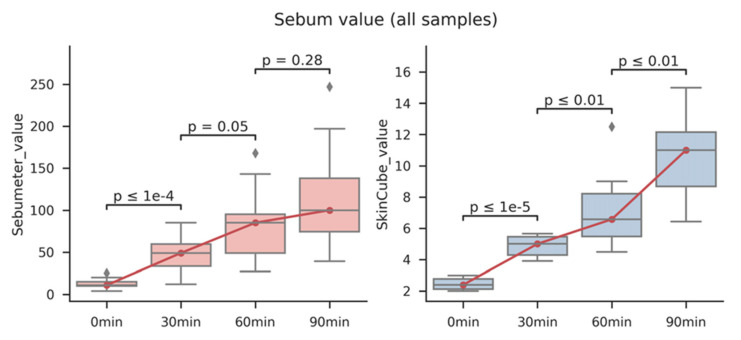
Comparison of Sebumeter^®^ and fabricated system using color sensor to detect skin casual sebum on sebum replacement time. Facial skin sebum of five participants was measured by sebum replacement time and compared using a Sebumeter^®^ and a fabricated system, respectively.

**Table 1 sensors-22-03016-t001:** Comparison of ORO-stained color intensity and intensity using the TCS 3200 color sensor on sebum replacement time.

Sample	Sebum Replace Time	Color Value Using Image J	R²
PM1	After washing 0 min	11.6995	0.9654
	30 min	19.791	
	60 min	35.778	
PM2	After washing 0 min	22.6905	0.9523
	30 min	26.699	
	60 min	35.7805	
PF1	After washing 0 min	29.9695	0.4083
	30 min	29.2205	
	60 min	31.35	
PF2	After washing 0 min	26.005	0.9789
	30 min	39.1375	
	60 min	46.9415	

**Table 2 sensors-22-03016-t002:** Comparison of ORO-stained color intensity and results of Sebumeter^®^ on sebum replacement time.

Sample	Sebum Replace Time	Color Value Using Image J	Sebumeter	Correlation
PM3	After washing 0 min	8.7	8	0.946
	30 min	47.2945	40	
	60 min	53.224	65	
PM4	After washing 0 min	10.8	68	0.996
	30 min	62.6	87	
	60 min	56.1	83	
PF3	After washing 0 min	10.8	31	0.994
	30 min	30.48	74	
	60 min	68.5	131	

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
