# Peer review of "A Real-Time Detection Device for the Rapid Quantification of Skin Casual Sebum Using the Oil Red O Staining Method"

_sensors, 2022, doi:10.3390/s22083016_

Round 1

Reviewer 1 Report

This manuscript described the development of devices for the detection of casual sebum based on ORO-staining. Some interesting results have been obtained. Howerer, some points are still necessary to be addressed.
1. The logical coherence, language grammar and punctuation of the article need to be revised.
2. In 1). Prepare the materials for collection of skin sebum part, polyvinyl chloride (PVC) and PVC films are the different materials or the same?
3. The paper and films cut to a 2 × 3 (cm) size areas were disinfected with alcohol. Alcohol belongs to an organic solvent. Whether it has an effect on the absorption of sebum? 
4. To improve comparability and clarity, the description of four commercial oil blotting papers need more data through the color sensor detection in different time. It's not just visual verification
5. In table1, PM1, PM2, PF1, PF2 represent which skin type. And from three participants? 
6. In table1, the R2 of PF1 was 0.4083. It may need more explanation on which factor causes it.
7. It need a table to compare with other detection method. 
8. What is the ultimate purpose/ practical application of this sensor on sebum detection. If for acne treatment, it may be need some sebum detection on patient.

9.The age range of participants appears to be wide. Because younger subjects may produce more oil than the older's.

10. From the table S2, R2 seems does not show a major advantage compared to sebumeter , although the deviation is smaller.

11. The fabricated color sensor-based device need more introduction about parameters and components.

Reviewer 2 Report

The article describes a technology for the detection of skin sebum using the oil red O staining method and an Arduino based device. The new tool was compared with Sebumeter®, a commonly employed technology for the quantification of skin sebum. The aim of the manuscript is providing a novel, cheap and reliable sensors for the detection and quantification of sebum, which is of high importance in the diagnosis and in the monitoring of different health conditions. Thus, the article falls within the topics of the MDPI-Sensors journal. However, there are important issues to be resolved prior to consider the work suitable for publication.

  • A deeper description of the methodology must be provided with more details. The manuscript lacks also in the full description of the results and the use of a communication instead of a classic manuscript must be carefully re-checked. A full manuscript will be surely more complete.
  • The introduction doesn’t not present the subject of the use of Arduino (or in general of open sources), while Arduino is a fundamental tool for the preparation of the device and is also reported as keywords. A suggested reference is D. Cressey, AGE OF THE ARDUINO, Nature 544(7648) (2017) 125-126.
  • There are many typos present in the paper.
  • 1-4 are quite outdated and being the topic quite important must be replaced with more recent ones.
  • The sketch of Arduino must be provided in the Supporting info, with more details of the device.
  • The Supporting info must be merged in a unique file.
  • The English of the manuscript is quite poor and should be substantially improved
  • Table 1 and 2 are not clear: what do PM and PF state for? Why PF1 has an R2 of 0.4083?
  • Figure 2b, error bars are not reported.
  • The novel deviced is described as cost-effective, but no analysis of the costs is provided. A LCC would be great.

Round 2

Reviewer 1 Report

The authors have addressed most of my questions.

Reviewer 2 Report

The authors well reviewed the paper. The aim of developing a reliable, low-cost and more importantly open source device for the quick quantification of skin sebum is highly relevant in the field on novel sensor. The article deserves now to be published.

I'd oly reccomend to revise a little more the supporting info and improve the presentation of the different figures and data in terms of mere representation aspect.